# Searching the web builds fuller picture of arachnid trade

Benjamin M. Marshall [1,2], Colin T. Strine [1,6], Caroline S. Fukushima[3], Pedro Cardoso[3], Michael C. Orr [4] & Alice C. Hughes [5,6✉]

Wildlife trade is a major driver of biodiversity loss, yet whilst the impacts of trade in some species are relatively well-known, some taxa, such as many invertebrates are often overlooked. Here we explore global patterns of trade in the arachnids, and detected 1,264 species from 66 families and 371 genera in trade. Trade in these groups exceeds millions of individuals, with 67% coming directly from the wild, and up to 99% of individuals in some genera. For popular taxa, such as tarantulas up to 50% are in trade, including 25% of species described since 2000. CITES only covers 30 (2%) of the species potentially traded. We mapped the percentage and number of species native to each country in trade. To enable sustainable trade, better data on species distributions and better conservation status assessments are needed. The disparity between trade data sources highlights the need to expand monitoring if impacts on wild populations are to be accurately gauged and the impacts of trade minimised.

[1] School of Biology, Institute of Science, Suranaree University of Technology, Nakhon Ratchasima, Thailand. [2] Biological and Environmental Sciences, University of Stirling, Stirling, UK. [3] Laboratory for Integrative Biodiversity Research (LIBRe), Finnish Museum of Natural History (LUOMUS), University of Helsinki, PO17 (Pohjoinen Rautatiekatu 13), 00014 Helsinki, Finland. [4] Beijing Institute of Zoology, Beijing, China. [5] Department of Biological Sciences, University of Hong Kong, Hong Kong, Hong Kong. [6] These authors contributed equally: Colin T. Strine, Alice C. Hughes. ✉email: ach_conservation2@hotmail.com

Wildlife trade is known to be among the major issues driving global biodiversity loss[1,2]. Among the many uses of wildlife, the pet trade represents a major component. Recent analysis has highlighted that pet trade represents a major potential threat to much greater numbers of species than previously realised, with 36% of reptiles and 17% of amphibians in trade, with around half individuals sourced from wild populations[3,4].

Terrestrial invertebrates represent a key component of the "exotic pet" trade[5], and understanding the dynamics of their trade requires greater exploration. Although evidence of widespread trade and even smuggling are known to impact potentially hundreds of invertebrate species[6], no global assessments of trade in invertebrate groups have been conducted. Whilst few studies exist on the potential impacts of invertebrate trade (e.g.[7]), popularity as pets or for specimen collectors is known to have nearly driven various species to extinction, especially where niche markets exist[8]. For the majority of invertebrate species, researchers lack precise information, despite potential declines, making further investigation an imperative[9,10].

Invertebrates are often neglected in conservation policy and practice due to biases in political, public and even scientific perceptions[11]. As a consequence, their conservation is chronically underfunded[12]. The Convention on International Trade in Endangered Species of Wild Fauna and Flora (CITES), the most relevant international convention that aims to protect traded species from overexploitation, lists a small fraction of terrestrial invertebrate taxa being traded and often threatened by trade[13]. This lack of knowledge on the population and range of invertebrates is evident; for example, of the over 1 million described invertebrate species[14], under 1% have been assessed by the International Union for Conservation of Nature (IUCN), and of these 28% remain Data Deficient (DD)[15]. This is actually an underestimation of the DD species, likely due to selective assessment of well-known individual taxa, as even well-studied groups like European bees show >50% data deficiency[16] and thus less studied invertebrate taxa are likely to be even less known.

Whilst overall invertebrates have been neglected by analysis on impacts of trade, the popularity of arachnids (particularly tarantulas and scorpions) combined with their extended lifespans and comparatively slow reproduction mean they may be less capable of withstanding unsustainable harvesting than some other arthropod taxa[17]. Yet arachnids have become popular pets, often regarded as "cool", and given that they require little space, arachnids make very practical pets for people without much space, such as many urban settings. Tarantulas and some butterflies are some of the only arthropod groups regulated by CITES, with listings often placed at the genus level (*Brachypelma* Simon 1891, *Poecilotheria* Simon 1885, *Bhutanitis* Atkinson 1873, *Ornithoptera* Boisduval 1832) to regulate trade by default for newly described species in some groups and minimising the risk that protected species are traded under the name of unlisted but related species hard to distinguish from the target species. Yet only 39 of the 52,060 described species of spiders[18] and only 1 of the 2348 scorpions (https://www.ntnu.no/ub/scorpion-files/index.php) is CITES listed[7], highlighting the inadequacy of relying solely on CITES data for understanding the true scale of wildlife trade for arachnids. The effectiveness of international regulation is challenging to assess since invertebrates are easy to launder under legally tradable species names, or to conceal due to their small size and difficulty of detection—unlike vertebrates, x-ray technology or thermal cameras cannot be used to enable detection for the majority of species. For arachnids, no global-scale assessments of species in trade have been conducted, despite the huge popularity of scorpions and various spiders exploited as pets. Recent studies show that in South Africa alone 132 species of tarantula were found in trade[19], highlighting the need for more exploration globally.

The lack of comprehensive IUCN and CITES listing and dynamics in arachnid taxonomy represents a challenge. Frequent changes in taxonomy and difficulty of identification by non-specialists mean that managing or even monitoring trade may be problematic. In addition, the arachnid groups Amblypygi (whip spiders), Uropygi and Opiliones (harvestman) are traded, but until recently there was no centralised species listing (until the launch of the World arachnid catalogue, which still lacks scorpions: https://wac.nmbe.ch/). This has made it difficult to follow the numerous taxonomic changes. Every year for the last two decades close to 1000 new species of spiders are described (ref. [18]; https://wsc.nmbe.ch/statistics/#). Similarly, 33 new Amblypygi species were described in a group of only 95 species in a single recent paper (e.g.,[19]). Many newly described arachnid species are likely to be endangered and potentially traded before or immediately after description[20]. Of those already described, only a fraction have been assessed by IUCN 0.65% spiders and 0.11% scorpions (318/49,170 spiders; scorpions 3/2763 (37 DD in total)).

The lack of basic data (such as life-history and ecological traits) and population trends for many species combined with the lack of centralised repositories of species names and synonyms in some groups (such as the scorpions), and a lack of consistent range information at resolutions below country or even biogeographical region, represent major barriers to understanding the dimensions and its impacts in trade in these groups. Another challenge to understanding the true dimensions of trade in arachnids is the small body size that facilitates smuggling and laundering (often via post or courier services), that pairs with low priority in enforcement, loopholes in environmental laws worldwide, and differing legislation[13,21,22] to reduce the seizures or even records in databases such as Law Enforcement Management Information System (LEMIS) from the US Fish and Wildlife Service. Combined, these factors may be contributing to an extensive web of under-regulated and unmonitored global arachnid trade.

## Results

**Overview**. All analyses included three major highly-traded arachnid groups, scorpions, spiders and whip scorpions. Though notably at least ten further amblypygids were included in listings on various sites as rapid rates of species description in a comparatively small group hindered useful keyword searching, so we instead used only the LEMIS database (hereafter LEMIS) for amblypygids. Our online search efforts of arachnid selling websites revealed a total of 1248 species (searching for current and historic synonyms and correcting to the currently accepted names; 1108 species detected with current species name matches), with a mean of 56.5 SE ± 7.83 species per website in the 2021 sample (range 1–491, 90/104 websites had species detected).

When combined with LEMIS and CITES trade databases, an overall total of 1264 species appeared in trade databases or on online arachnid selling websites. Of the 1264 species in trade, 78.5% (993 species) were only detected online; 15 species were unique to LEMIS (of 267 LEMIS total), and no species was unique to the CITES trade database out of the 30 species listed the CITES trade database (Fig. 1).

The 1264 species detected in trade included: 11 (8.0%) Uropygi species, 350 (12.7%) scorpions, and 903 (1.8%) spiders (Fig. 1). In terms of genera, this included five Uropygi, 70 scorpions and 296 spiders, giving a total of 371 genera. When only genera which had species in trade were considered, 18% of scorpions, 9% of spiders and 11% of Uropygi are in trade.

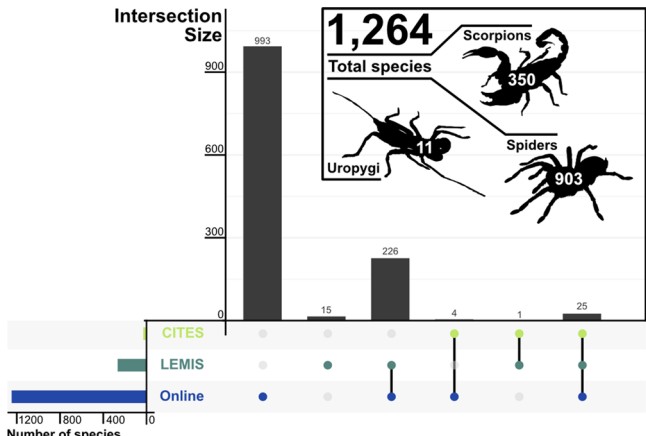

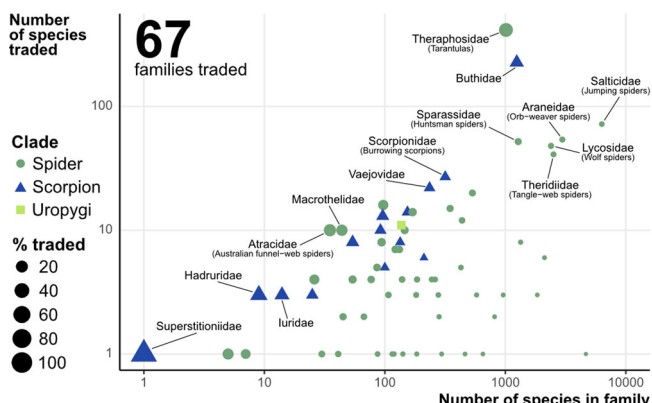

**Fig. 1 UpSet plot showing how the three different sources overlapped.** UpSet plot of how the three different sources overlapped in terms of species, with an insert showing how the total number of species was split between three major clades. Lower left hand bar chart shows the number of species detected via each source.

**Fig. 2 Number of species plotted against the number of species traded.** Each point represents one family, with some of the larger families named. Highlighted with text are some of the highest percentage traded families, with common names if applicable. n.b. log scales both *x* and *y*.

The few species detected in all three sources comprise 2% (25/1264) of the total (though this is in large part because of how few species are appearing in the CITES trade database), highlighting the need to sample widely to fully characterise the arachnid trade at the species level. In addition, we found at least 95 potential species in trade under common names which noted the locality or colour patterns and included sp. and the descriptor rather than an accepted species name (Data S9 and S10), and in some cases even noted that it was likely an undescribed species.

**Extent of trade per group**. Considering the three datasets, a total of 66 families and 371 genera were in trade; however, the percentage traded for each family and genus varied dramatically (Fig. 2). Unsurprisingly, many small genera (i.e., with low diversity; Fig. 2) had a high percentage of species in trade, but some of the large groups such as the tarantulas have over 50% species traded (Fig. 2). For genera with at least one species traded, an average of 31.7% of species per genus of scorpions was in trade, 37% of spiders and 28.5% of Uropygi. If we remove genera where very few species are in trade (for example over 10 species but with only a single species in trade), the percentage of species in trade for these more popular groups goes to 33% for scorpions,

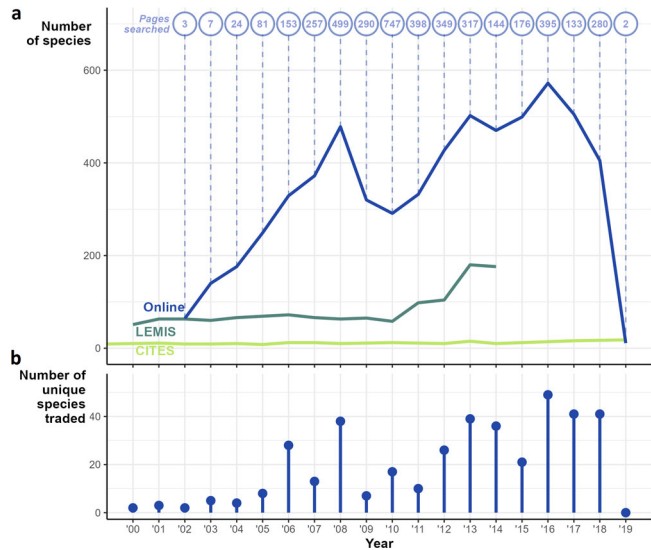

**Fig. 3 Numbers of species listed per year from the three data sources.** **a** The raw number of species in each of the data sources, supplemented with a count of the archived pages searched to describe the varying online sampling effort. **b** The number of species unique to that particular year, from any source.

44.2% for spiders and remaining at 28.5% for Uropygi. This highlights that for taxa which are popular in trade, high percentages of species per genus may be traded. At the family level, we found that over 415 species (41%) of Theraphosidae tarantulas and 227 species (18.25%) of buthid scorpion are in trade, though smaller families often show substantial percentages of species in trade (Fig. 2).

**Assessments and vulnerability**. The majority of arachnid species have not been evaluated by the IUCN and therefore have no Redlist status: 99.34% of spider species, 100% of Uropygyi and 99.9% of scorpions have no assessments in IUCN. Among those that have IUCN assessments, there are no Uropygyi species listed (but two Amblypygi), only three scorpions, and many spider species. There are 56 spider species listed as Critically Endangered (2 in trade-3.5%), 69 Endangered (6 in trade-8.7%), 53 as Vulnerable (4 in trade-7.4%), and 35 as DD (5 in trade-14.3%, though given how few species are evaluated this designation is not very informative; Supplementary Fig. S1; IUCN assessments as of 2021-09-15). Despite having many species evaluated in the IUCN threat categories, only a fraction of arachnid species are regulated by CITES: (36 spider species, from one family, and 3 species of a single genus of scorpion). Thus, less than 1% of the species of each group are specifically regulated by an international trade agreement.

**Trends over time**. Notably, with the exception of CITES (which has a less dynamic listing process, monitoring few species and therefore cannot be considered representative), both LEMIS and online forums indicate an increase in the live trade of arachnids, with almost 600 species listed as for sale on one site in 2016 (Fig. 3a). It should be noted that decreases after 2016 for online trade are likely a result of fewer available pages from the archive at the time of sampling, rather than fewer species in trade (and overall trend in online trade numbers should be treated with caution due to the sensitivity to sampling). Each year included unique species not seen in other years (e.g., observed just in 1 year), peaking at 40 unique species in 2016 (Fig. 3b). Species considered rare and those that are rare in trade may be listed

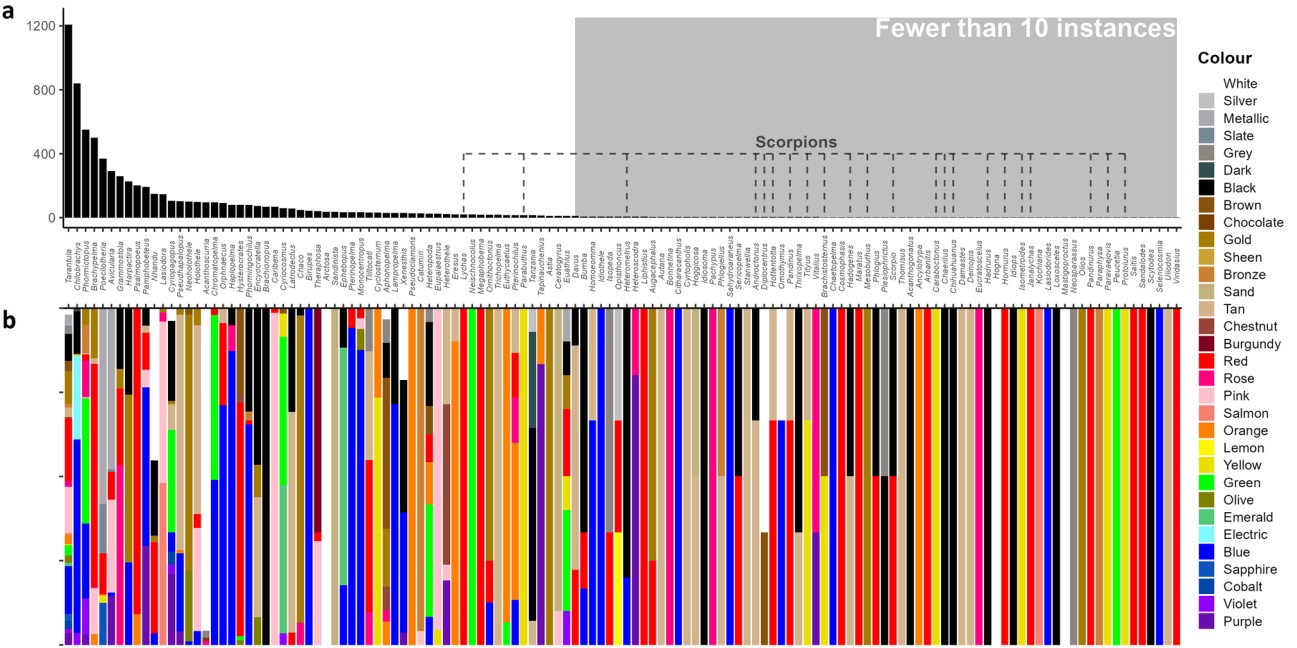

**Fig. 4 Count and colours linked to mentions of arachnid genera. a** Raw counts of the number of times a genus keyword appeared alongside a mention of colour (based on actual colour listings). Most genera mentioned are spiders, and scorpions are highlighted with grey dashed lines. **b** The proportion of genera keyword detections linked to a particular colour, with legend detailing the colour word detected.

more infrequently, and therefore missed in snapshots further highlighting the risk of under-estimating species traded if reliant upon snapshot sampling.

**Language assessments.** While we used systematic searches to identify arachnid selling websites in nine languages, no Portuguese sites yielded any keyword hits (possibly partly due to a suboptimal search string and the rigorous legislation regarding the commerce and possession of native and exotic species). Via our ad hoc sampling, we also detected a Swedish site. Comparing languages, English had the highest number of species with over 600 species, of which 228 were unique, but also had the greatest sampling effort (Supplementary Fig. S2). This was followed by German, with 508 species in trade, of which around a fifth (108) were unique. Other languages had smaller numbers, but most still contained unique species, highlighting potentially different market preferences in different regions.

**Colour and ranges.** Novelty and uniqueness may be major drivers in the trade of some groups, especially in some tarantulas and in the Salticidae (jumping spiders). Visual inspections of sites showed that place names (e.g., South Mindanao (in the Philippines), Kaeng Krachan (in Thailand)) were often appended to scientific names, as was colour (often used instead of a species name to note a distinct colour morph). In some instances, mentions of colour or locality may indicate hidden diversity, as yet undescribed species which have already entered trade. Our assessment of colour co-occurring with generic names revealed 32 colours or colour descriptors associated with genus names, with a mean of 2.48 SE ± 0.263 colours per genus keyword. We did not use traits from species, only listing colours explicitly noted in proximity to the genus name on a web page, as this may reference hidden diversity (Fig. 4a).

The word "tarantula" had the highest association with a colour (i.e., listed in association with colour within a few words of the genus name), with over 1,200 mentions of a colour alongside the genus name (Fig. 4a). We found 25 different colours associated

with tarantula either as part of the common name, or indicating undescribed diversity as the consistent use of colloquial names by collectors may represent (Fig. 4b). This was followed by *Chilobrachys* with 840 listings, then *Phormictopus* at 551. Most species with colours associated were spiders (largely tarantulas), with some spider groups such as Salticidae jumping spiders and scorpions also listed at lower levels. Readers should note that colour descriptions may not be indicative of particular variants or morphs; for example, frequent mentions of red or pink can be tied to use of the common names of Tliltocatl/*Brachypelma* spp. (e.g. redknee tarantulas), and *Avicularia* spp. (e.g., pinktoed tarantulas); however, colour morphs could be potentially undescribed species in some cases (Data S10).

**Quest for novelty and newly described species.** Like many invertebrates, high rates of species description are associated with most arachnid groups[22]. Many species have been described since 1999: including 1265 species of scorpion (with 54 detected in trade; 4.3% of newly described species), 19 species of uropygid (with 1 in trade; 5.3%), and 15,790 of spiders (with 123 in trade; 0.8%). Within spiders, of the 15,790 post-1999 species, 358 were tarantulas who appear comparatively more frequently traded (86 in trade; 24%).

In total, 17,074 arachnid species from our studied groups have been described since 1999, of which 178 (1.04%) have since appeared in trade, including 25 species which appeared in trade within 1 year of description, and 14 species apparently in the same year as description (Fig. 5). Although many species were only detected in the wider website "snapshot" of 2021, they are likely to have been traded earlier but simply not detected. Excluding those detected in only the snapshot online data, average lag time is ~4.55 SD ± 4.31 years: 4.36 SD ± 4.52 for spiders, 4.90 SD ± 3.94 for scorpions (no post-2000 described Uropygi were detected outside of snapshot data). However, whilst the overall rates of description have been high, the patterns for different taxa varies (e.g., Buthidae: 660 species described post-1999 species, 38 traded, 5.8%; Theraphosidae: 358 post-1999 species, 86 traded, 24%).

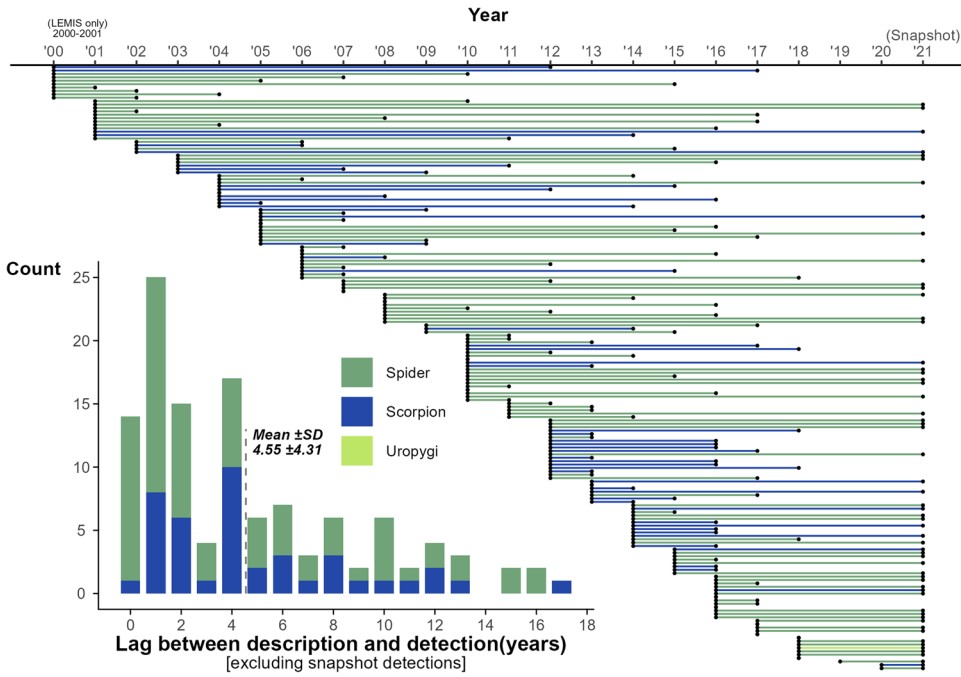

**Fig. 5 Years of detection in the trade of species described post-1999 (via LEMIS or online).** Inserted histogram shows the count of the year lags for those species detection could be connected to an exact year (i.e., not detected in the 2021 snapshot sampling).

**Origins and sources**. Our explorations into origins of exports warrants caution as it only refers to a subset of species listed within LEMIS (Fig. 1), as websites do not mandate information on origin or source (though some state if individuals are captive bred or wild sourced, this is likely biased towards reporting captive origin for various reasons such as disease risk, perceived impacts, legal regulations). However, as the European market is often stated as less regulated than that of the United States (largely in the fora of websites) it is likely that similar or greater percentages of individuals from the European market are also sourced from the wild.

Only 0.034% of the individuals recorded in LEMIS were seized, and the vast majority of individuals are reportedly serving commercial purposes (91.8%; 4,265,900/4,647,476). Overall, the percentage of individuals sourced from the wild is around 70.4% (3,270,299 individuals), whereas only 28.5% are stated to come from captive sources.

These ratios vary dramatically between groups. The most traded group noted in LEMIS is *Pandinus* Thorell 1876 (Emperor Scorpions), which includes over 1 million individuals, of which 77.43% were wild caught; similarly, 600,000 individuals belonging to the genus *Grammostola* Simon 1892 were imported, and 88.9% came from the wild, possibly in violating of sanctions[23]. Three of the four other families detected having over 100,000 individuals imported showed 77.8–93.6% of individuals coming from the wild. Only specimens of the genus *Mesobuthus* Vachon 1950 were detected largely coming from captive sources (89.2%; Supplementary Fig. S3). Regarding families and genera traded at slightly lower levels, only a minority were largely sourced from captive sources. For taxa traded at 100,000–10,000 individuals (Supplementary Fig. S4), only 6 of the 23 groups came from captive sources (mean 67% from wild, and up to 99.6% for some groups). The percentage of individuals sourced from the wild decreases slightly for genera traded at a slightly lower level: at 61% for 10,000–1000 (Supplementary Fig. S5), and at 68.2% for those traded at 1000–100 individuals with the majority of groups being sourced entirely from the wild (Supplementary Fig. S6), and a similar trend with a mean of 68.2% for those traded at below 100

individuals in LEMIS (Supplementary Fig. S7). Notably, CITES-listed genera such as *Brachypelma* and *Poecilotheria* were largely recorded being imported as captive bred individuals, although some were still recorded as wild caught. The majority of other genera were largely recorded as coming from the wild.

If we considered the individuals coming from the wild (via the LEMIS database), some patterns emerge. Three countries were detected as the main commercial source for the raw number of individuals imported into the USA (Fig. 6a): Ghana (713,870 of 722,685 individuals wild sourced: 98.8%), Chile (621,413 of 688,756 individuals wild sourced: 90.2%), and China (449,712 of 1,182,441 individuals wild sourced: much lower than others at 38.0%). Considering the number of genera traded, China (28 genera, 2nd most) and Chile (23 genera, 7th most) have notably high numbers of wild-sourced genera traded, although these numbers include non-native species (e.g., for example 50.6% of individuals exported from Chile were not classed as native). Germany joins China with 28 genera, and both are only surpassed by Tanzania's 50 genera (Fig. 6b). The countries considered hotspots in terms of quantity (Fig. 6a, b) are not the same as the ones in terms of percentage of individuals coming from the wild, these mostly found in Europe and in North America (Fig. 6c).

In addition, based on the LEMIS data, we examined species listed as exported from the wild, and analysed if they were coming from countries listed as native vs. non-native (highlighting either false/misleading reporting or laundering, though knowing true ranges of these species can be challenging, and some species may have been re-exported). For countries with over 1000 exported wild individuals, 19 countries have at least 50% of "wild" individuals coming from countries they are not native to, with 8 showing 100% (including Vietnam with 73,315 individuals and Nicaragua at 13,081 individuals) and a further four at over 90% (including Honduras with 21,415 individuals, Malaysia at 38,096 and Guatemala at 40,750). Other countries also have large numbers, for example Tanzania has 51.5% of "wild" individuals (27,658) which are not native, as well as 89,459 individuals (50.6%) in Chile, if these typify general trends this suggests very high levels of exports from this region.

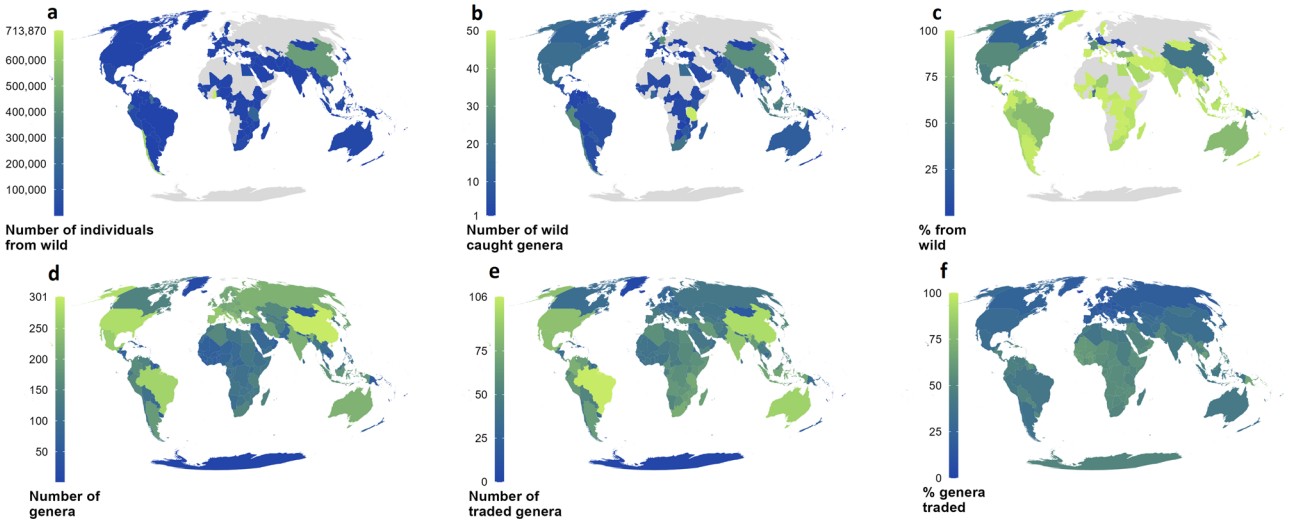

**Fig. 6 The source of traded spiders and scorpions harvested from the wild. a–c** show the source based on the origin listed in the LEMIS database. **d–f** show the natural distributions of all traded spiders and scorpions based on all data sources, number of species traded are shown in supplements. N.B., The natural distributions are only accurate to the country level, and in some cases represent extrapolation from broadly defined distributions (e.g., a "South American" distribution would be mapped as including all countries in South America).

Out of the 28 species which had over 1000 individuals in trade, with >50% of wild caught individuals from countries for which they are not listed to be native in, these five have over 10,000 individuals purported to come from the wild from countries they are not listed as native in: *Heterometrus spinifer* (Ehrenberg 1828) (75,060), *Grammostola spathulata* (F. O. Pickard-Cambridge 1897) (notably a synonym of *Grammostola rosea* (Walckenaer 1837)) (72,096), *Aphonopelma seemanni* (F. O. Pickard-Cambridge 1897) (69,413), *Haplopelma lividum* (Smith 1996) (26,470) notably a nomenclature change from *Cyriopagopus lividus* (Smith 1996), *Ephebopus murinus* (Walckenaer 1837) (11,465). This also suggests that synonyms may be being used to misrepresent the origins of species in trade, for particularly popular species.

We mapped richness and trade patterns for spiders and scorpions at species and genus level. For spiders, the digitisation of species ranges from the World Spider Catalog (WSC)[18] website yielded 134,187 connections between species and national areas for spiders alone. Richness peaked in China with 5139 species, followed by Brazil (3972), Australia (3906), and the United States (3880) also showing high richness (Supplementary Fig. S8). Despite smaller areas, Mexico (2466) and South Africa (2055) had very high richness. The highest percent of species in trade come from various islands and Cambodia, but parts of the Middle East, Uruguay and Suriname also show 10–15% of species are traded (Supplementary Fig. S8). However, if genera with two species or fewer in trade are removed the percentage of species of trade in remaining groups is up to 60%, with high levels also found in Bolivia and across much of North Africa, and still including 113 species being traded from Brazil.

When examined at the genus level, patterns in genera roughly follow these species-level patterns, with China (301), the USA (279), and Brazil (267) hosting the greatest number of genera (Fig. 6d). The number of genera in trade varies significantly between countries, with the highest number in trade coming from Brazil with at least 106 genera in trade, as well as high numbers in North America and Australia (Fig. 6e). The highest percentage of genera in trade come from various island states, as well as Cambodia, and a number of West African states (Fig. 6f).

If we examine some of the most traded genera or families of spider, many regions trade 100% of their species from groups in

trade (e.g., for some genera, every species from the genera in the region was in trade), this includes theraphosids with 415 species in trade in total (41%) and up to 88 species in trade from countries such as Brazil, and the majority of species of native tarantula across most of Africa in trade, as well as most small islands. Patterns of exploitation varied between the most traded groups, with for example 40% of Theridiidae spp. from any given country in trade (particularly in Africa), up to 100% of Sparassidae spp. in trade (largely in Europe), 25% of Lycosidae spp. in trade (Middle East) and 50% of Araneidae spp. and 35% of Salticidae spp. in trade from parts of Africa and the Middle East (Supplementary Fig. S9).

For scorpions, richness peaked in Mexico with 301 species, with high levels of richness also seen in Brazil (181), the United States (140) and India (137). Patterns of trade also follow these general patterns, with the greatest number coming from Mexico (28) and South Africa (24). In terms of the percentage of species, however, whilst South Africa was moderately high at 21%, many countries were higher. Jordan trades 42% of species, Botswana 38%, Mauritania and Egypt both export 37% of species, and a further eight African countries at least 30% of species, whereas outside Africa (and parts of the Mediterranean) few countries have over 20% of species in trade (Supplementary Fig. S10).

## Discussion

Whilst wildlife trade is now often acknowledged to be a major driver of global biodiversity losses, the potential for trade to threaten the survival of various terrestrial invertebrate species has been largely overlooked, even by regulatory frameworks such as CITES[2]. Yet, some taxa (such as some arachnids) share many of the traits known in other taxa to be associated with vulnerability, such as long lifespans (exceeding 30 years in some taxa), and their popularity as pets is increasing without parallel conservation management strategies of trade for most species[23]. In total, we detected 1264 species in trade. Our searches for arachnid species differed markedly from our previous searches on reptiles and amphibians where LEMIS showed similar numbers of species to online search efforts. For arachnids, a startling 73.8% (993) of species were only for sale online and not listed in trade by either LEMIS or CITES. This likely stems from both a lack of regulation and ability to send "slings"/spiderlings as well as adults through

postal services (details of this are featured on online trading forum, and YouTube). Increasing popularity of arachnid keeping, or selling online, is suggested by the number of species for sale on a single site (Terraristika) being only 150 in 2003, yet increasing to almost 600 by 2016 (although note the variation in sampling effort), and often showing five times the number of species traded according to LEMIS in any given year. It is also important to note that quantitative analysis of individuals in trade was limited to imports via LEMIS. Google trends data (searched 2021-11-04: https://trends.google.com/trends/explore?date=2016-06-06%202021-06-06&q=%2Fg%2F120jx064,%2Fm%2F01phsx,%2Fm%2F0755b&hl=en-US&tz=-480; Data S14) also shows peaks in interest coinciding with periods of lockdown in the spring and autumn of 2020, meaning more species and individuals may have entered trade over that period. The format of websites prevented accurate quantification of the number of individuals traded: websites may only provide a basic stocklist, readvertise individuals across multiple pages, host fraudulent adverts, or not disclose the number of individuals available, as understanding the dimensions of trade based on online data is challenging and more standardised approaches are needed[24].

It is also notable that many species are currently considered country endemic (76.4% of all arachnid species range data indicates single-country endemism, 94% are in under five countries Supplementary Fig. S11), with 814 species with a range of just one country in trade (736 exact match). This indicates potential vulnerability to trade, although present distributions are likely often incomplete as data for most invertebrates extremely scarce, the so-called Wallacean shortfall[25]. This highlights the need for more national level action to both collate data and develop policies to improve protection of potentially threatened invertebrates like many arachnids, and to include such data in National Biodiversity Strategy and Action Plans (NBSAPs) as well as developing better policies for the export of these species. Assessment of origins was especially difficult, with distributions often limited to country or regional resolutions, highlighting that many species are very range limited and thus highly susceptible to over-collection. Our maps highlight several possible key hotspots, but any more detailed assessment of origin, or predictive modelling will require far more consistent and higher quality spatial descriptions of species' distributions, including better efforts to digitise data collected by scientists in research of these species in these native ranges to provide base map data[25–27].

The coverage through international bodies tasked with regulating or monitoring trade in these groups (CITES and IUCN) appears lacking, not only failing to include over 99% of species, but even in groups which have some level of CITES coverage (such as various tarantulas). Almost half scientifically described tarantula species are traded, and almost 20% of scorpions assigned to the Buthidae. Notably, over 25% of tarantulas described since 2000 are already in trade, and a lack of regulations mean that even undescribed species can be exported without oversight[28]. Despite this lack of coverage, even for species evaluated as threatened by the IUCN Redlist, over a third of the "randomly selected" subset of species in trade are threatened, with around a quarter of listed species shown to be Endangered, Critically Endangered or even DD, yet they can still be exported subject to minimal regulation, and are often unregulated domestically as they are rarely covered by domestic policy. The lack of baseline data on species distributions means that understanding the impacts on wild populations is almost impossible, as highlighted by the large number of the DD listed species that are in trade, despite the small number of arachnids listed by the IUCN as DD. Additionally, it is notable that for the subset of species included in LEMIS up to 99% of individuals in some of the most traded genera come from the wild (such as emperor

scorpions, with over one million individuals imported into the US alone). Interestingly this volume of trade is likely to have occurred in violation of CITES (the EU had already suspended imports from some countries since 2008), resulting in suspensions for four African countries in 2020 (Benin, Ghana, Liberia and Togo) and Guinea in 2021[29], highlighting the lack of enforcement of bans on trade in invertebrates.

Whilst the majority of groups with available data come from the wild (few online advertisements stated origin and source and so assessment of wild imported individuals was limited to LEMIS), many individuals stated to come from the wild are in fact listed as being exported from countries where they are not native. This may be because of a lack of comprehensive data on species distributions, but it may also highlight possible issues with reporting, or laundering through countries with fewer regulations, again highlighting the need for better regulations on trade and export. Various countries in both Africa and South America are centres for the export of wild-sourced individuals, notably Chile and Ghana. Yet over 50% of individuals of species listed as exported from Chile are not considered to be native, suggesting it may offer a conduit for species exported from other parts of South America. In addition, some of the species traded in the largest volumes from "the wild" from countries they are not listed as native to are tarantula species, highlighting the need for additional regulation and monitoring in the trade of the group, and to better track and monitor the original sources of re-exported individuals[30]. Furthermore, many species within LEMIS still use junior synonyms, masking the true volumes of trade for certain groups.

The challenging, and often changing taxonomy of these groups causes additional problems[31], as demand for novelty and rapid rates of species description in the group may be spurring a drive for new species, with at least 189 species described since 2000 already traded, including at least 25% of all new tarantula species. The likely lag between formal species description and the trade of "variants" mean that rare forms may be traded before a new species is formally accepted[31]. These unusual or novel taxa or morphs may be traded under a geographic label (e.g., Kaeng Krachan, South Mindanao) or a colour, with the colour or place appended to of up to 100 species or genera in advertisements for tarantulas (Data S10). Former studies also highlight the trade in undescribed species[32], and the high undescribed diversity within groups like the tarantulas is also well established[20]. Our analysis highlights that many groups are vulnerable to trade, yet a lack of regulation, as well as fluid taxonomy and a lack of data on most species[33], form barriers to monitoring in the group. It is also notable that former research shows that species are commonly named and advertised correctly, but that cryptic diversity within "species" may exist, and in some cases represent several species i.e., species complexes[20], which aligns with the collector value of novelty and diversity within the group. For instance, the very popular, jet-black *Grammostola pulchra* Mello-Leitão 1921 has many close relatives of similar coloration that cannot easily be distinguished, and there is a possibility that *Grammostola quirogai* Montes de Oca, L., D'Elía, G., Pérez-Miles 2015 is also traded under the name *G. pulchra* as the two are challenging to distinguish without high levels of expertise[34]. These issues are further highlighted if we look at species lists from individual collectors on forums (Data S10); for example, a single collector had 205 species of arachnid, including 16 species with cf. or sp. (which can both denote that the putative species has not been formally described) then a locality, five further with a colour or pattern (diamondback), seven further without the species identified, and one with three different colour morphs (*Dolichothele diamantinensis* Revollo, Silva and Bertani 2017 (Green/Blue/Black). For a collector with this quantity of species, it is highly likely that some of

these represent as yet undescribed species. In fact, if we collate such references, at least 100 species listings on one online forum have such notation (sp. then a colour or locality), highlighting the need for further work to describe species, as well as clearly identifying those in trade to ensure trade is not impacting what may be regionally endemic species which are already in trade[31,35]. Based on the number of recently described species in trade, as well as the number of species in trade under colloquial or specialist (non-scientific) names, there is the potential for novelty to be a key driver in the arachnid trade. Junior synonyms can also mean that a species which is subject to CITES or other regulations may continue to be traded under a junior synonym or old name, and thus newer legislation is likely not to be applied when the species is traded, hindering trade regulation and accurate monitoring of trade volumes. The presence of potentially undescribed species, and the high number of species with large numbers of individuals listed as wild caught yet exported from countries where those species are not native to highlights the lack of effective regulation for the majority of species. In some regions the potential for invasions is a risk to native taxa, as high levels of trade occur in regions like South Africa which is likely to be suitable for many species[6].

Analysis here was limited to a subset of languages and generally based on websites, but we observed that trade also takes place via social media (e.g., Instagram and Facebook). However, as species names may be embedded in photographs, and names which may not match (requiring external validation, and such sites frequently require membership or involve trades via private messaging), the number of species in trade is likely to be substantially higher than detected within this study. We also only searched online shops and stocklists, largely ones that did not require membership or verification.

Analysis in more challenging taxonomic groups (whip scorpions and whip spiders) was particularly difficult due to a lack of any centralised taxonomic backbone and regular updates in the species listed within such groups[19]. For example, whilst we noted at least 15 amblypygids for sale and that high numbers within the group are imported via LEMIS, we did not undertake a comprehensive search due to high rates of species description, and high numbers of recently described species[19]. While partially overcome in this assessment of arachnids, factors such as private social media trading, fluid taxonomy[20], rapid rates of species description, and decentralised taxonomic species lists (until very recently) combined may represent an insurmountable barrier to such assessments within other invertebrate groups.

Varying legality can lead to different levels of trade detectability depending on the country, and transit routes. For example, online trade of wild animals online is illegal in Portugal (Lei no.95/2017) and we were unable to find any open shops or stocklists in Portuguese. We are aware that in some countries this is likely severely limited by the efficacy of search-terms. Former searches (reptiles, amphibians) of the internet based on a subset of keywords were effective[3,4], here many sites were only found through the forums of existing web-stores as many sites discovered via web searches contained non-arachnid products. This suggests that keywords on many of these sites are not as readily accessible to Google and other search-engines, and thus many more species may exist in other online platforms. In addition, other uses such as consumption of some large spider species as food, and regional exports in regions like Asia, may mean overall numbers of arachnids traded greatly exceed what we detected with this methodology. Consumption is also an overlooked factor, as are standards to assay where individuals have been collected in the wild from and what impact it may have on wild populations[36,37]. Anecdotal reports (and observations from the authors) highlight the sale of likely trafficked spiders for consumption in night

markets across parts of South China (likely from neighbouring Lao and Vietnam and Cambodia, as supplies have declined dramatically with border restrictions during the pandemic), and trade in newly described local species (*Chilobrachys lubricus* Kun et al., 2021) is also recorded[38]. There is almost no systemic accessible information about such trade, in recent years non-traditional markets in the developed world have also started importing arachnids as novelty food[39,40]. Thus, despite the high levels of trade in certain groups, particularly buthid scorpions and tarantulas, even higher numbers are likely to be in trade, unnoticed, unmonitored and almost entirely unrestricted.

The majority of individuals in trade for the vast majority of species (for which data exists) still come from the wild; the lack of over-arching regulation means that developing, monitoring and enforcing these harvest quotas and origins remain virtually unknown. Even for the small proportion of species within LEMIS, many species are still either imported as junior synonyms or from countries where they are not native, highlighting the lack of any effective regulation for most of the group. Re-exporting individuals without clear information on trade routes can make it very difficult to assess what is in trade, and what impact it may have on wild populations. Better agents and standards are urgently needed, as even those making efforts to follow legal and ethical standards still source almost all their imports from the wild. Given the lack of population or accurate distribution data for most species, the precise impacts are impossible to assess, and few species have the data needed for adequate assessments, highlighting that legality and sustainability cannot be conflated. Existing regulations in most countries do not provide sufficient safeguards for most species and perpetuate an idea that "legal trade is sustainable," when there are currently insufficient data to link the two. The impact of collection (or even the volumes) cannot be calculated due to the tiny proportion with barely adequate monitoring. Huge volumes of seizures in some countries such as the Philippines:[41–43] where trade is more regulated also highlights the high volumes of trade which may be occurring without oversight, and similar efforts in other countries may also reveal untold volumes of internal trade.

Another limit of our assessment is the lack of quantitative data for the majority of species, as individuals in trade may have no data on volumes, or may be re-listed, making assessments of their quantities impossible. A number of sites sell "mystery boxes" of spiders (we did not systematically search for the frequency of mystery box listings), with limited information on what they may contain. The fact that they can exist highlights the lack of regulatory oversight for most species, not only is it impossible to monitor what is traded, but they could be used to facilitate laundering of protected species. Making trade sustainable would require a regulatory system to monitor both what is in trade, and where it is coming from, as well as population data on wild populations. Yet such measures would require an entirely different approach to the trade in wildlife, with regulatory systems in place to facilitate and document the export of live animals, and the verification and registration of captive bred individuals for any international export. To ease the regulatory burden of trade, such strictures (facilitate regional trade of captive bred individuals, and heighten regulations of external trade) could pertain to trade-blocs/areas and apply solely to international sale, to complement the existing CITES system.

Whilst attempts to restrict trade are sometimes stated to contravene sustainable livelihood provision, unsustainable trade cannot provide stable economic gains in the long-term, and ultimately undermines future access to that same "livelihood" to other species which rely on them, and the ecosystem services they provide. Regulating trade to sustainable levels, including assessments and monitoring within National Biodiversity status and

action plans (NBSAPs) to enable quotas of wild capture to be set on such a basis is in everyone's interest, as is a transition to certified breeding facilities to guarantee quality and minimise wild-collection[21]. Certification could be made contingent with certain regulations which includes registration of new hatches, and would thereby enable a transparent way to monitor trade and only allow collection based on clear non-detriment findings (e.g., in CITES). Understanding if trade is sustainable not only requires an understanding of the volume of trade and the source of individuals collected as well as the size of the wild population and capacity to recover from harvest. Yet it should be noted that even if good regulations exist, how effective they are is determined on what enforcement actions are implemented, and these remain a challenge[21].

DNA barcoding is becoming a more standard means of identifying vertebrate species, such a practice is more challenging with invertebrates. Removing adequate biomass for barcoding (i.e., removing a leg, see[44,45] may need to be replaced with feeding them set foods and then sequencing individual DNA from faeces, or using exuviae (moults) to correctly identify species using barcoding and without harm to the animals[46]. Applying such techniques to verify species identification prior to trade (after collection, prior to sale or export) would overcome at least one of the barriers to making trade sustainable by providing a means to reliably identify species. Interestingly UNCTAD (which regulates commodity trade) has started to discuss better monitoring methods of the trade of wildlife, and their existing national and international species for the trade of livestock can provide examples of regulatory systems could help monitoring wildlife trade[47]. Furthermore, a high proportion of arachnid trade is likely illegal, and better understanding and regulating trade will require better awareness of the impacts of trade, and better regulation which reflects how species are traded (such as postal services) in addition to modes to verify species identity and reduce laundering species under false names or source codes. Ultimately, until better data and regulations exist, there will remain no way to assay the impacts of trade on the majority of species. However, what is clear is that many individuals do come from the wild, and that many species are regional or even single-country endemics, and it is likely undescribed species are traded under colloquial names whilst remaining undescribed. It is also key to note that many arachnids show traits associated with vulnerability to unsustainable harvest in other taxa, meaning that many species of spiders and scorpion may be scuttling towards extinction, without the data to tackle both illegal and unsustainable trade, as much legal trade is likely to be unsustainable.

Whilst many people assume that invertebrate trade could not pose a risk to species survival, we show that, from what data are available, over 50% species within popular large families may be in trade, and that for those with import data a high percentage still come from the wild. We cannot quantify the impact of wild capture on these species, but the lack of global monitoring is cause for concern. CITES regulates trade for under 1% of species and under 0.034% of individuals in LEMIS were shown to originate through seizures, suggesting that illegal import is likely to be either coming via other means (such as postal services) or being laundered to avoid detection, though it may also be that these taxa are viewed as "less important" and less oversight and so seizure data may not be representative of what is being traded[48]. Many species are likely endemic to a single-country, but a lack of data on most species ranges means assessing vulnerability and developing appropriate management or conservation policies is currently impossible. These species are undoubtedly vulnerable to unsustainable trade, especially as novelty appears poised to play a role with colour, colloquial names, and place of origin listed alongside for sale arachnids online. In any event, since arachnids often show high levels of endemism, efforts are needed to monitor what is in trade, to verify identities, and trace origins of specimens belonging to this group in order to prevent potentially unsustainable trade and species extinction arising from trade without the data or regulations needed to ensure sustainability.

## Methods
Our online sampling methods largely follow protocols detailed in[3,4], though we limited our online searches to online shops and did not extend to social media. Large portions of code are directly re-used from those papers, although we provide modified code with this paper additionally. For keyword searches and data review we used R v.4.1.1[49] via RStudio v.1.4.1103[50], and made wide use of functions supplied by the anytime v.0.3.9[51], assertthat v.0.2.1[52], dplyr v.1.0.7[53], glue v.1.4.2[54], lazyeval v.0.2.2[55], lubridate v.1.7.10[56], magrittr v.2.0.1[57], 17urr v.0.3.4[58], reshape2 v.1.4.4[59], stringr v.1.4.0[60], and tidyr v.1.1.3[61] other specific package uses are listed during the methods description. We used the grateful v.0.0.3[62] package to generate citations for all R packages. Code and data used to produce figures and summary data are also available at: 10.5281/zenodo.5758541.

**Website sampling and search**. We searched for contemporary arachnid selling websites using the Google search engine and targeted nine languages (English, French, Spanish, German, Portuguese, Japanese, Czech, Polish, Russian). Terms were created to be inclusive, so only spiders and scorpions were on the initial search string as specialist groups may exist for either, but are unlikely for smaller arachnid groups, which were often listed under "other" in online shops. Terms were selected to be encompassing so that any sites listing variants of "spider" or mentioning arachnid in the chosen language were selected. Whilst some groups such as tarantulas are more popular as pets such sites will not omit translations of spider and should also be captured in the search, hence Terraristika (as was shown in previous analysis of amphibians and reptiles) listed the greatest number of species, despite not being a specialist site. We used the localised versions of each of these languages with the following Boolean search strings:

- English: (Spider OR scorpion OR arachnid) AND for sale
- French: (Araignée OR scorpion OR arachnide) AND à vendre
- Spanish: (Araña OR escorpión OR arácnido) AND en venta
- German: (Arachnoid OR Spinne OR Skorpion OR Spinnentier) AND zum Verkauf
- Portuguese: (Aranha OR escorpião OR aracnídeo) AND à venda
- Japanese: (クモ OR サソリ OR クモ型類) AND (中村彰宏 OR 販売)
- Czech: (Pavouk OR Štír OR pavoukovec) AND prodej
- Polish: (Pająk OR Skorpion OR pajęczak) AND sprzedaż
- Russian: Продажа пауков OR скорпионов

We undertook these searches in a private window in the Firefox v.92.0.1 browser[63] to limit the impacts of search history. These keywords were used to identify sites which may be selling arachnids, which could then be checked before a comprehensive scrape.

For each language, we downloaded the first 15 pages of results between 2021-06-06 and 2021-07-07 (or fewer in the result that the search returned fewer than 15 pages: German 8 pages and Spanish 14 pages). This resulted in ~1270 sites that could potentially be selling arachnids. After removing duplicates and simplifying the URLs (so all ended in.com,.org,. co.uk etc.; Code S1), we reviewed each site for the following criteria (2021-07-31 to 2021-08-02): whether they sell arachnids, the type of site (trade or classified ads), the order arachnids were listed in (e.g., date or alphabetical), the best search method for gather species appearances (see below for hierarchical search methods), a refined target URL listing species inventory, the number of pages within the website potentially required to cycle through, and if the search method required a crawl, whether the site explicitly forbade crawling data collection and whether we could limit the crawl's scope with a filter on downstream URLs. Finally, we assigned all suitable sites with a unique ID. We have made a censored version of the website review results available in Data S1. In addition to the systematic search for arachnid trade, we added 43 websites discovered ad hoc from links on previously discovered sites (many listed online shops), those listed in other journal articles on invertebrate trade (i.e.,[6]) or from discussion with informed colleagues (between 2021-08-07 and 2021-09-15). After reviewing these ad hoc sites (2021-08-07 to 2021-09-15), we had a combined total of 111 sites to attempt to search for the appearance of arachnid species.

Our searches of websites took one of five forms (Code S2), designed to minimise server load and limit the number of irrelevant pages searched, while ensuring we captured the pages listing species. We prioritised using the lowest/simplest search method possible for each site.

**Single page or PDF**. For websites that listed their entire arachnid stock on a single page, we retrieved that single page using the downloader v.0.4 package[64]. In cases where the inventory was listed in a PDF, we manually downloaded the PDF and used pdftools v.3.0.1[65] to assess the text.

**Cycle**. Some websites had large stocklists split across multiple pages that could be accessed sequentially. In these cases, we used the downloader v.0.4 package[64] to retrieve each page, as we cycled from page 1 to the terminal page identified during the website review stage. Two sites required a slight modification to the page cycling process: as the sequential pages were not defined by pages, but by the number of adverts displayed. In these instances, we cycled through all adverts 20 adverts at a time (i.e., matching the default number displayed at a time by the site). For all cycling we implemented a 10 s cooldown between requests to limit server load.

**Level 1 crawl**. For websites that split their stock between multiple pages, but with no sequential ordering, we used a level 1 crawl, via the Rcrawler v.0.1.9.1 package[66] to access them all. For example, where a site had an "arachnid for sale" page, but full species names existed only in linked pages (e.g., "tarantulas", "scorpions").

**Cycle and level 1 crawl**. Some websites required a combined approach, where stock was split sequentially across pages, and the species identities (i.e., scientific names) required accessing the pages linked to from the sequential pages. In these cases, we ran the initial sequential sampling followed by a level 1 crawl.

**Level 2 crawl**. Where level 1 crawls were insufficient to cover all species sold on a site, we used a level 2 crawl to reach all pages listing species. This tended to be the case on websites with multiple categories to classify and split their stock (e.g., "arachnid"—"spider"—"tarantula").

For all crawls, we used a cooldown of 20 s between requests to limit server load, and where possible we limited the scope of the crawl (i.e., linked pages to be retrieved) using a key phrase common to all stock listing pages (e.g., "/category=arachnid/").

In addition to the sampling of contemporary sites, we explored the archived pages available for https://www.terraristik.com via the Internet Archive (2002–2019[67]). Terraristika had been previously shown as a major contributor to traded species lists[4], and the website's age and accessibility via the internet archive meant it was one of the few websites where temporal sampling was feasible. We used pages retrieved via the Internet Archive's Wayback machine API[68], via code created for[3,4]. The code used was based on the wayback v.0.4.0 package[69], but additionally made httr v.1.4.2[70], jsonlite v.1.7.2[71], downloader v.0.4[64], lubridate v.1.7.10[56], and tibble v.3.1.3 packages[72] (Code S3).

**Keyword generation**. We relied on multiple sources to build a list of arachnid species (spiders, scorpions and uropygi). For spiders we used the WSC (ref. [18]; https://wsc.nmbe.ch/dataresources; accessed 2021-09-18). We filtered the WSC dataset to remove subspecies, then used a combination of rvest v.1.0.1[73], dplyr v.1.0.7[53], and stringr v.1.4.0 packages[60] (see Code S4) to query the online version of the WSC database to retrieve all synonyms for each species. Where the synonyms were listed with an abbreviated genus, we replace the abbreviation with the first instance of a genus that matched the first letter of the abbreviation.

We combined the WSC data with a list manually retrieved from the Scorpion Files[74] (https://www.ntnu.no/ub/scorpion-files/index.php; accessed 2021-09-19). For the uropygi species, we combined species listings from Integrated Taxonomic Information System (ITIS[75]; https://www.itis.gov/servlet/SingleRpt/RefRpt?search_type=source&search_id=source_id&search_id_value=1209 and https://www.itis.gov/servlet/SingleRpt/SingleRpt?search_topic=TSN&anchorLocation=SubordinateTaxa&credibilitySort=TWG%20standards%20met&rankName=ALL&search_value=82710&print_version=SCR&source=from_print#SubordinateTaxa; accessed 2021-09-19) and the Western Australian Museum[76] (http://www.museum.wa.gov.au/catalogues-beta/browse/uropygi; accessed 2021-09-19). We were unable to source reliable data on all scorpion and uropygi synonyms; therefore, we used all names listed from all sources, but made note of those names considered *nomen dubium*. Our final keyword list contained 52,111 species, 94,184 different species names, with mean of 1.81 SE ± 0.01 terms per species (range 1–41). For summaries of total species, we relied on the species classed as accepted by the species databases (WSC, Scorpion Files, ITIS and the Western Australian Museum).

**Keyword search**. We successfully retrieved 3020 pages from 103 websites (mean = 28.78 SE ± 11.42, range: 1–1077), and used a further 4668 previously archived pages. To prepare each of the retrieved web pages for keyword searching, we removed all double spaces, html elements, and non-alpha-numeric characters, replacing them with single spaces (Code S5). For this process we used rvest v.1.0.1[73], XML v.3.99.0.8[77], and xml2 v.1.3.2[78] packages. This process increased the chances that genus and species epithets would appear in a compatible format when compared to our keyword list. The process was not able to repair abbreviated genera, or aid detection where genus and species epithet were not reported side-by-side.

Due to the large number of species we were forced to adapt previous searching methods, instead implementing a hierarchical genus-species search (Code S6). We searched each retrieved page for any mention of genera, then only searched for species that were contained within that genus. We did not differentiate whether the genus was currently accepted or old, so if a species had ever belonged to a genus it

was included in the second stage of the search. The specifics of the keyword search used case-insensitive fixed string matching (via the stringr v.1.4.0 package[60]). While collation string matching would have helped detect species with differently coded ligatures or diacritic marks, the occurrence of ligature and diacritic marks are infrequent in scientific names and did not warrant the considerably increased computational costs.

Via the keyword search we recorded all instances of genus matches, species matches, the website ID, and the page number. We also collected the words surrounding a genus match (3 prior and four after) as a means of exploring common terms that may be used to describe the genera.

We provide the compiled outputs from searching contemporary and historic pages in Data S2–S4. Prior to combining these two datasets into a final list of traded species, and summarising the overall patterns, we cleaned out instances of spurious genera and species detections. Predominantly this included short genera names that could appear at the start of longer words (e.g., terms such as: "rufus", "Dia", "Diana", "Mala", "Inca", "Pero", "May", "Janus", "Yukon", "Lucia", "Zora", "Beata", "Neon", "Prima", "Meta", "Patri", "Enna", "Maso", "Mica", "Perro"; we already implemented a filter that required genera to be preceded by a space and thus these were not part of the species name). We are confident these genera should be excluded, as none had species detected within them.

**Trade database and third-party data**. We downloaded United States Fish and Wildlife Service's LEMIS data compiled by[79,80] from https://doi.org/10.5281/zenodo.3565869 (Data S5). We filtered the LEMIS data to records where the class was listed as Arachnida (Code S6).

We downloaded the Gross imports data from the CITES trade database from the website and filtered to Class Arachnid, years 1975–2021[81] (accessed 2021-09-15; Data S6), and downloaded the CITES appendices filtered to arachnids[82] (Data S7).

We downloaded the IUCN Redlist assessments for arachnids from https://www.iucnredlist.org[83] (accessed 2021-09-15; Data S8).

**Species summary and visualisation**. We compiled all sources of trade data (online, LEMIS, CITES) into a single dataset detailing which genera/species had been detected in each source (Data S9 and Code S7). We used two criteria to determine detection, whether there was an exact match with an accepted genus/species or whether there was a match to any historically used genera/species name. Because of splits in genera, the "ANY genera" matching is likely overly generous. For broad summaries we rely on the "ANY species" name matching.

We used cowplot v.1.1.1[84], ggplot2 v.3.3.5[85], ggpubr v.0.4.0[86], ggtext v.0.1.1[87], scales v.1.1.1[88], scico v.1.2.0[89], and UpSetR v.1.4.0[90] to generate summary visuals (Code S8; Code S9). We added additional details to the upset plot and modified the position of plot labels using Affinity Designer v.1.10.3[91]. We also used Affinity Designer to create the Uropygid silhouette for Fig. 1. We obtained public domain licensed spider and scorpion silhouettes from http://phylopic.org/ (https://phylopic.org/image/d7a80fdc0-311f-4bc5-b4fc-1a45f4206d27/; http://phylopic.org/image/4133ae32-753e-49eb-bd31-50c67634aca1/).

**Descriptions and colours**. We explored the lag time between species descriptions, and their detection in LEMIS or online trade (Code S10). We relied on the description dates provided alongside the lists of species names. Unlike the broader summaries, we restricted explorations of lag times to species detected only via exact matches (operating under the assumption that newly described species traded swiftly after description would be using the modern accepted name). We distinguished between those species detected only in the complementary data, as the earliest trade date was not known; therefore, our summaries of lag time are based on those species detected in a particular year either via LEMIS or temporal online trade.

Following a visual inspection of sites where we often noticed listings with either colour or localities (e.g., "Chilobrachys spp. "Electric Blue" 0.1.3. Chilobrachys sp. "Kaeng Krachan" 0.1.0. Chilobrachys spp. "Prachuap Khiri Khan": Data S9). We explored the words that surrounded detected genera. After using the forcats v.0.5.1[92], stringr v.1.4.0[60], and tidytext v.0.3.1[93] package to compile common terms and remove English stop words, we determine colour was frequently mentioned (Code S11). To filter out non-colour words, we used wikipedia's list of colours (https://en.wikipedia.org/wiki/List_of_colors:_N%E2%80%93Z). Once cleaned, we further removed terms that are ambiguously colour related (e.g., "space", "racing", "photo", "boy", "bean", "blaze", "jungle", "mountain", "dune", "web", "colour", "rainforest", "tree", "sea"). We then summarised this data as the counts of instances where a genus appeared alongside a given colour term (n.b., counts are therefore impacted by any underlying imbalances in how many times a site mentioned a genus). We plotted all colours using the same hex codes listed on the wikipedia page, with the exception of "cobalt", "grey", "metallic", "slate", "electric", "dark", "sheen", and "chocolate" that required manual linking to a hex code.

**Summary of trade numbers**. We summarised LEMIS data using a number of filters (Code S12). Following[3,4,94], we limited our summaries to items that feasibly can be considered to represent whole individuals (LEMIS code = Dead animal BOD, live eggs (EGL), dead specimen (DEA), live specimen (LIV), specimen (SPE),

whole skin (SKI), entire animal trophy (TRO)). We describe the portion of trade that is prevented (i.e., seized, where disposition == "S"). We classed non-commercial trade as anything listed as for Biomedical research (M), Scientific (S), or Reintroduction/introduction into the wild (Y). For captive vs. wild summaries, we treated all Animals bred in captivity (C and F), Commercially bred (D), and Specimens originating from a ranching operation (R) as originating from captivity. We only included animals listed as Specimens taken from the wild (W) in wild counts. The few instances that fell outside of our defined captive vs. wild categorisation are treated as other. For summaries of wild capture per genus, we relied entirely on LEMIS's listings of genera, making no effort to determine synonymisations. We did filter out those listed only as "Non-CITES entry" or NA. We used the countrycode v.1.3.0[95] package to help plot the LEMIS countries of origin. Taxonomy represents an ongoing challenge, we were limited to recognising the species listed in the aforementioned databases, generating synonym lists from these sources, and attempting to reconcile these lists. Rapid rates of species description means that compiling comprehensive lists can be challenging, and species may be traded under junior synonyms or old names, and newer descriptions may not have been added to sites[96]. We were also limited to platforms that advertised using text not images, as images can be challenging to identify accurately.

**Mapping**. Mapping species is challenging due to the lack of standardised data on species distributions. Spider distributions were mapped based on the data in the World Spider Catalogue (Data S12). Firstly, the localities associated with each species were collated into four spreadsheets based on the data provided in the WSC (WSC[18]; https://wsc.nmbe.ch/dataresources; accessed 2021-09-18), these listed (1) country, (2) region, (3) "to" (where the range was given as one country to another) and (4) Island.

Before processing any "introduced" localities were removed, the four sheets were then checked for any simple spelling errors (in islands file) or mislistings (i.e., regions in the islands file). Country data were cross-referenced with the names of country provided by Thematic Mapper to standardise them (https://thematicmapping.org/; Data S11). This was done by uploading data into Arcmap and using joins and connects to connect it to the standard country name file, and any which could not be paired were corrected to ensure all could be successfully digitised.

Regions were digitised based on accepted names of different regions and included 33 different regions (see supplements) for each of these the standard accepted area within each of these regions was searched online to determine the accepted boundaries. These were then selected from the Thematic mapper, exported and labelled with the corresponding region. Once this was completed for all 33 regions they were merged and exported to a geodatabase. The spreadsheet listing regional preferences of each species was also uploaded to Arcmap 10.3, then exported into the geodatabase, then connected to a regional map using joins and relates to connect the regional preferences from the spreadsheet to the shapefiles. The new dbf was then exported to provide a listing of each species and each country in the region it was connected to, and then copied into the same csv as the corrected country listings.

For preferences listed as "to" we first separated each country listed in the "to" listings into a separate column, then developed a list of species and each of the countries listed in the "to" list (which frequently between 5–6). These were then corrected to the standard names from thematic mapper for both countries and the regions used in the previous section. We then merged the countries and regions file and added fields of geometry in ArcMap to provide a centroid for each designated area. This table was then exported and joined and connected to the species in the "to" file. This data was then converted to point form and turned to a point file, then a minimum convex polygon (convex hulls) developed for each species to connect the regions between all those listed. These species specific minimum convex polygons were then intersected with the countries from Thematic mapper, and then dissolve was used to form a shapefile that just listed species and all the countries between those ranges. This was then exported and merged with the listings from countries and regions.

The islands file included both independent islands (which needed names corrected, or archipelago names given) and those that fall within a national designation. For those islands we replaced the island name with that of the country, as listings of species may be particularly poor, and tiny non-independent islands are not visible in the global-scale analysis.

This forth database table was then merged with the former three, and remove duplicates used to remove any duplicate entries, as species often had individual countries listed in additions to regions or "to". This was then uploaded into Arcmap and exported to a geodatabase file then connected to the original Thematic mapper file and exported to the geodatabase to yield 134,187 connections between species and countries. This was then connected to our main analysis to include the trade status, and CITES and IUCN Redlist status for each species for further analysis.

Scorpion data was considerably messier than that on the world spider catalogue. Firstly, we downloaded all scorpion data from iNaturalist and GBIF[97,98] (search; scorpions), removed duplicates, then cross-referenced these with the thematic mapper file within Quantum GIS. Species listed in regions where they were clearly not native (i.e., a species listed in the UK when the rest of that species or genus were in Australia) were removed, and all extinct species were excluded.

In addition, all the "update files" were downloaded from the "Scorpion files", the PDFs collated then using smallpdf tools the tables were extracted into excel form and cleaned to include just species and country listing. This was added to the countries listed for species within[99] and[100] though this was restricted to a subset of species. The data were all collated into an excel file with the species name, and country listing. This was then added to all the data from https://scorpiones.pl/maps/. These maps have a good coverage of species countries, but are apparently no longer being updated (Jan Ove Rein pers comm 2021) hence the need for further data to provide complete and updated and comprehensive coverage for all species. Country names were then standardised based on the Thematic Mapper standards (Data S13 and Data S11). Species names were then cross-referenced to those listed in the Scorpion files, any not matching were checked as synonyms and converted to the accepted name (though the only collated data for Scorpion synonyms was on French-language Wikipedia, i.e., see https://fr.wikipedia.org/wiki/Bothriurus). Once all country and species names were corrected this provided a listing of 4059 species-country associations. These were then associated with country files in the same way as spiders. We plotted spider and scorpion species/genera, as well as LEMIS origins, using ggplot2[85], combining Thematic world border data (https://thematicmapping.org/) with summaries of species/genera/and trade levels. Species listed in a single-country (and thus more likely to be country endemic) were also counted using summary statistics, so that species most vulnerable to trade could be noted separately.

**Reporting summary**. Further information on research design is available in the Nature Research Reporting Summary linked to this article.

## Data availability

All data are available as supplements to this manuscript to allow full reproducibility of results. Data used to produce figures and summary data are also available at: https://doi.org/10.5281/zenodo.5758541.

## Code availability

Code used is available at https://doi.org/10.5281/zenodo.5758541. Throughout the methods we have indicated the stage of analysis each data component was used and the code script connected.

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

## Acknowledgements
We thank Pavel Torovov, Akihiro Nakamura, Inês Silva, and Ross Creelman for their help translating search phrases. We thank Chen Yanhua for their efforts reviewing website content. We thank the Suranaree University of Technology Institute of Research and Development and School of Biology for providing the resources required to undertake this research.

## Author contributions
Conceptualisation—ACH, CTS, and BMM. Data curation—BMM and ACH. Formal analysis—BMM and ACH. Investigation—BMM and ACH. Methodology—BMM and ACH. Supervision—ACH and CTS. Visualisation—BMM and ACH. Writing—original draft—ACH and BMM. Writing—review and editing—ACH, BMM, CTS, CSF, MCO, and PC.

## Competing interests
The authors declare no competing interests.
