## [Peer Review File · Communications Biology]

Reviewers' comments:

Reviewer #1 (Remarks to the Author):

The study, "Searching the web builds fuller picture of arachnid trade. Unravelling the worldwide web of arachnid trade" impressively describes the situation of the international trade in terrestrial invertebrates, especially Arachnida, using the best available methods. Very comprehensive with many numerical examples, it is made clear how much international monitoring is lagging behind in making this trade transparent and sustainable.

Right at the beginning of the manuscript I was struck by the enormous abundance of figures (which are of course indispensable), but therefore the abstract must be well structured. To my mind, it reads a little difficult, so see my few ideas as a comment in the manuscript.

I have made further comments in the manuscript. Essentially, the clarity with which specific facts are described is sometimes lacking, see my comments on this in the MS.

In particular, I would also like to point out the mention of some taxonomic issues. With regard to taxonomy, a section is missing in the methods that provides information on which sources were used, also with regard to the very dynamic situation of new descriptions and the accompanying taxonomic adjustments, including the recognition of species complexes.

A work that is often cited is that of Zhang et al. (2011) Animal biodiversity: An outline of higher-level classification and survey of taxonomic richness. *Zootaxa* 3148: 1-237." Then some of the main papers on the taxa studied, ie. Araneae, Scorpiones and Uropygi should be mentioned.

At times I have pointed out the lack of citations, which would be appropriate at the end of some statements. An additional reference would be e.g:

<https://www.science.org/doi/abs/10.1126/science.363.6430.914>.

In the text, I also noted twice that the problematic issue of exporting *Pandinus imperator* from West Africa, in terms of quotas, wild capture and ranching, could be elaborated a little.

Overall, this study is very comprehensive in its methodological approach and, with the data generated, can certainly be regarded as basic work from which further studies can be generated. Overall I suggest minor revision.

Reviewer #2 (Remarks to the Author):

The authors present research into the global arachnid trade. They look at online trade sites, US imports, and the CITES trade database. They characterize the trade of arachnids by species perform summary analyses about: taxonomic trends, conservation (IUCN) characteristics, year of species discovery vs. entry in trade, country of origin, and more.

It is my assessment that the authors present a valuable contribution that is methodological sound and robust. Their methodology has been used in prior research and is well-documented in the literature. Their results are novel and illuminating on a topic that, to date, has not received much attention. Further, their discussion of results is well presented and not overstated – all the necessary caveats are discussed and woven throughout the paper. Importantly, specific conservation implications are not discussed because their results do not merit such a deep dive, but the authors point to what research needs to be done in order to have specific conservation recommendations. My review is short because I believe this paper is worthy of publication as is. I have read in the manuscript in full and given it careful consideration. I have left some minor comments below that the authors may wish to consider.

I do not wish to remain anonymous and have signed my review.
Oliver C. Stringham

oliver.stringham@adelaide.edu.au

Minor comments:

Line 59: can you give another example other than the Hitler's beetle? Apparently the species was literally named after Hitler, so I don't endorse the common name and would prefer it not to be used where possible. Or perhaps a species of blind cave beetle and leave it at that.

Introduction in general: can you briefly state why arachnids are being traded? Or at least what is known about that? It's not until the discussion that pet trade and consummation are mentioned, but I think it's worth a mention in the introduction.

Line 135: saying only 2% seems a bit misleading because the maximum number all 3 sources can have in common is 30 (the data source w/least # spp. = # cites spp.), if that makes sense? I would re word to reflect this.

Line 137: 'around 100 potential species' – just say the exact number

Line 148: why the world 'could', is it because the true number of species in the group is not known? It wasn't clear (to me) from the text.

Line 153-154: is this sentence just to caveat you didn't sample every seller/location/etc on earth? To me, it seems out of place or missing a reason why you are cautioning that the number of species in trade may be higher than you observed.

Line 163: what does N.b. mean? Is this a journal specific thing?

Figure 2: Please mention that one data point represents one taxonomic family. It is evident but it took me a little bit to figure out.

Line 181 – 184: I don't really see that post-2016 trend in the data. 2017 had 500 spp in 133 pages (3.8 spp per page), 2018 had ~400 spp in 280 pages (1.4 spp per page), which seem like a similar rate to the other years. One way to tell for sure is to plot the # spp./page for each year. Further, the number of unique spp for '17 and '18 are higher than most other years. For those 2 reasons, the caveat in these lines seems out of place/wrong. Please reconsider this statement.

Line 217: the 'or potentially indicating' clause doesn't make sense to me. What is potentially indicating undescribed diversity?

Figure 4B: Are these colors the literal color descriptions used by sellers? Or was some aggregation used?

Lines 231 – 234: what to the percentages indicate?

Line 240: I think standard deviation is a better metric for variation. Looking at Figure 5, the variation around the mean does not look like 0.4 years – I understand that may be the SE but to me that's not really useful.

Figure 5: can I suggest a discrete color ramp. They tend to me more understandable. In ggplot2 it would be `scale_fill_fermenter()`.

Figure 5: I think the color ramp limits should be the same between D and E. That way they can be directly compared? Just a suggestion.

Figure 5f: Are there arachnids in Antarctica?

Figure 5b&c: Why does the US show up as having arachnids traded. I was under the impression you were solely looking at imports to the US. But perhaps you are also looking at exports out of US? If this is the case, please state this in the main text.

Line 299: Also important to note that these individuals could have been 're exported' meaning they

entered the non-native country and then were exported at a later time after spending some time in the country. Or are captive bred and then exported as wild caught, etc. We talk about this a little in: Sinclair et al., The International Vertebrate Pet Trade Network and Insights from US Imports of Exotic Pets, *BioScience*, Volume 71, Issue 9, September 2021, Pages 977–990, <https://doi.org/10.1093/biosci/biab056>

Line 362: any references for this? Or was this something you observed in the advertisements you saw? Please specify.

Line 369: The link did not work for me. Perhaps worth taking a screen shot and putting in Appendix so that there already be a record of what you are referring to.

Line 375: we discuss the different types of websites and the limitations of web data for wildlife trade, if it is a useful reference here, feel free to use it: Stringham, et al. (2021), A guide to using the internet to monitor and quantify the wildlife trade. *Conservation Biology*, 35: 1130-1139. <https://doi.org/10.1111/cobi.13675>

line 399: CITES doesn't handle domestic trade. I don't think you were saying this here but it was a little ambiguous (in my opinion)

line 416: we noted this in the above-mention Sinclair paper for vertebrates so it's a problem not just limited to arachnids. Perhaps a wildlife/pet trade problem in general

line 439: what does cf. mean? Sorry for my ignorance but I haven't come across this

lines 449 to 451: very nicely worded

line 498: it is unclear (to me) how the use of junior synonyms results in not effective regulation. I don't get the link. Perhaps ineffective enforcement, if you believe the use of junior synonyms means that the true species won't get flagged. I think the US system should (although I don't really know) have a cross reference to taxonomic database. Either way please make this link explicit because right now it doesn't sit right.

Line 512-514: I think you are reading too much into these mystery boxes. First, these boxes would only be at the seller level, not at the import/export level, right? So presumably the actual species/individuals will still be subject to country-border level regulations. Second, I don't think the mystery boxes necessarily mean anything bad. It could mean store just wanted to get rid of common stock. This might be an optimistic take but I don't see the benefit to sell desired/rare species this way because they would get more money if they sold them normally. Unless you know more specifics on it or have a reference, I would remove this part.

Line 520: This sentence is unclear to me. What exactly is a trade-blocs/area in this context? A country?

Line 531: It's worth noting that many of these regulation suggestions require effective enforcement. And that enforcement is difficult. This is touched on Cardoso's and Fukushima's recent synthesis papers.

Line 541: what are these lessons? Left me on a cliff-hanger. I think it's worth mentioning one of them.

Line 562: Lack of seizures doesn't mean that illegal species aren't passing through USFWS surveillance. We observed they do occasionally let 'illegal' species in without seizing them. For example, we noted this for an Australian endemic reptile: Heinrich, et al. (2022), Strengthening protection of endemic wildlife threatened by the international pet trade: The case of the Australian shingleback lizard. *Anim. Conserv.*, 25: 91-100. <https://doi.org/10.1111/acv.12721> . I would guess this is even more of a problem for invertebrate if it's a problem for a highly recognizable vertebrate species.

Line 572: I know our papers came out around the same time but the above Stringham et al. (2021) outlines a procedure for monitoring the web for wildlife trade. If you feel it's appropriate, you can cite it here as well.

Methods: worth mentioning you did not look at social media/social media groups.

Great paper, well done!

Reviewer #3 (Remarks to the Author):

I found this manuscript well written and the analyses well done. The actual content of the paper is novel, well researched and greatly brings to the forefront the rapidly growing plight in world trade of targeted invertebrates.

My only criticism would be that I would have liked to see 'Discussion' follow 'Methods', rather than the way the paper is presently laid out. I think categories like 'Conclusion' or the final 'Discussion' should follow all the methods that were used to reach those categories.

The remainder of my comments that need attention are found in the returned ms, however, once addresses, they should not impede the publication of this ms.

Reviewers' comments:

Reviewer #1 (Remarks to the Author):

The study, "Searching the web builds fuller picture of arachnid trade. Unravelling the worldwide web of arachnid trade" impressively describes the situation of the international trade in terrestrial invertebrates, especially Arachnida, using the best available methods. Very comprehensive with many numerical examples, it is made clear how much international monitoring is lagging behind in making this trade transparent and sustainable.

Right at the beginning of the manuscript I was struck by the enormous abundance of figures (which are of course indispensable), but therefore the abstract must be well structured. To my mind, it reads a little difficult, so see my few ideas as a comment in the manuscript.

Response: Thank you, we have restructured the abstract to hopefully make it more accessible and easy to follow for all levels of expertise

I have made further comments in the manuscript. Essentially, the clarity with which specific facts are described is sometimes lacking, see my comments on this in the MS.

In particular, I would also like to point out the mention of some taxonomic issues. With regard to taxonomy, a section is missing in the methods that provides information on which sources were used, also with regard to the very dynamic situation of new descriptions and the accompanying taxonomic adjustments, including the recognition of species complexes.

A work that is often cited is that of Zhang et al. (2011) Animal biodiversity: An outline of higher-level classification and survey of taxonomic richness. *Zootaxa* 3148: 1-237." Then some of the main papers on the taxa studied, ie. Araneae, Scorpiones and Uropygi should be mentioned.

Response: Thanks, this has now been noted in the methods. We have also clarified that we relied on the "accepted/valid" species as listed on the given species database (WSC, Scorpion Files, ITIS and the Western Australian Museum). This was part of the keyword compiling/generation process.

At times I have pointed out the lack of citations, which would be appropriate at the end of some statements. An additional reference would be e.g:

<https://www.science.org/doi/abs/10.1126/science.363.6430.914>.

Response: Thank you! Added

In the text, I also noted twice that the problematic issue of exporting *Pandinus imperator* from West Africa, in terms of quotas, wild capture and ranching, could be elaborated a little.

Response: Thank you, this is fascinating, and we have elaborated on this

Overall, this study is very comprehensive in its methodological approach and, with the data generated, can certainly be regarded as basic work from which further studies can be generated. Overall I suggest minor revision.

Response: Thank you, we appreciate that and the useful comments throughout
Manuscript comments

I would reword the abstract partially more concise, probalby restructure and first present the total number of species found in trade, then separate etween CITES and non CITES species and then add infomation on threatened species, undescribed species and taxonomic uncertainty. To evalaute

would be to add info how trade could affect endemism, as most species are native to one country, as I understand.

Response: restructured to try to make the text accessible, but also more concise

I would then also use "whip scorpions" here, as you also did not refer to "araneae" or "scorpiones"

Response: Done

included in title > delete

Response: Done

Amphibian-amphibian

Response: Done

may be redundant

Response: removed

sourced from wild populations (reference?)

Response: We moved the references to the end of the sentence and edited as suggested

probably< best already refer to "terrestrial invertebrates" here

Response: edited

add 1-2 references

e.g. MUAFOR FJ , LEVANG P, ANGWAFO TE, LE GALL P (2012) Making a living with forest insects: beetles as an income source in Southwest Cameroon. International Forestry Review 14(4): 1-12.

or examples of the bird wing butterfly trade:

Schütz, P. (2000). Flügel hinter Glas – Der Insektenhandel in Deutschland unter besonderer Berücksichtigung der Schmetterlinge (Lepidoptera). TRAFFIC-Europe/Um- weltstiftung WWF-Deutschland. https://www.traffic.org/site/assets/files/5620/flugel_hinter_glas_der_insektenhandel_in_deutschland_unter_besonderer_beruecksichtigung_der_schmetterlinge.pdf

Response: Thank you, we've added Muafor et al., 2012.

of terrestrial invertebrate taxa

Response: edited

species or genera,- change to taxa

Response: edited

Add exploited

Response: Added

probably here write, "dynamics in taxonomics" as below you have changes in taxonomy already

Response: changed to "dynamics in arachnid taxonomy"

would also write out english names when scientific names of orders are mentioned for the first time, as here the "whip spiders" or Opiliones the "harvestmen"

Response: Edited as suggested

Uropygids....

Response: Changed to whip scorpions

the LEMIS database

Response: Changed

Edit sentence order

Response: Changed

of the 30 species listed on the appendices of CITES.

Response: Changed

this needs to be reworded, e.g., when only taxa to genus level were considered???

Response: rephrased. As not all genera have any species in trade, we calculated an average overall, and for the genera with at least some trade

would delete this, a subjective evaluation

Response: we removed 'only'

can this be specified?

Response: we added the exact number and refer to the supplement

would swap these two, first family then genus

Response: done

perhaps better, "small-sized genera"

Response: Added "(i.e. with low diversity)"

Species

Response: changed

Change groups to taxa

Response: changed

over 415 tarantula species (Theraphosidae spp.) (41% ..."

Response: changed

and 227 scorpion species (Buthidae spp.) ..

Response: edited

capitalize "Luridae"

Response: It reads Iuridae, which is correct (though very hard to tell in this font and whilst capitalized)

i assume here is meant, species that are only listed in the categories LC and DD? I would then rather write, species not evaluated in a threat category incl. NT.

However, if it is meant that species not evaluated in the IUCN Red List, then write this.

Response: we mean that very few arachnids have any assessment by the IUCN, and have now tried to make that clearer

but this addition is redundant when comparing with the previous line.

Also, make sure when to use acronyms for the first time, in order to then only be able to refer to them in the further course of the text.

Response: this has been made clearer, and the acronym removed as redundant

Despite having many species evaluated in the IUCN threat categories

Response: edited

a tiny proportion

Response: edited

regulated under CITES, i.e. 36 spider ... genus of scorpion." REST > delete

Response: thanks for noting this, corrected

" in the live trade of arachnids (or at ..."

Response: edited

Why is that?

Response: The limited and varying web pages per year is due to the pages available in via the Internet Archive. The pages available via the Internet Archive's wayback machine are supplied by various "crawling" efforts. We are not aware of anything describing the crawling processes nor the balance between the difference crawling sources (i.e., organisations collecting and searching web pages). There is also an option for users to request a page be archived. In this manuscript we simply aimed to maximise our sampling, as we had no basis to correct this unknown archiving sampling process.

New?

Response: only found in one year, now noted in text

Species considered rare ...

Response: corrected

would refer to the countries here, Philippines and Thailand I assume

Response: correct-added

characteristic colour morph of the respective species?

Response: yes, made clearer

Not italic!

Response: corrected

??? something missing here?

Response: thanks, yes-the sentence has been adjusted

probably not readable

Response: We have cleaned up the figure to increase the axis text.

is "Search for new species descriptions" meant here, or rather "Quest for novelty and new scientific descriptions" ?

Response: changed to "Quest for novelty and newly described species"

I would distinguish between geographic origin and collecting source (wild, captive, ranch) and therefore separate that other section with a new heading.

Response: we added "and sources" to the title to better reflect both elements

see quotas for the West African states at "speciesplus" and potentially CITES notifications for discussion

Response: thank you, we now reference this, a very useful dimension to add

they were-removed

Response: removed

how is this meant?

Response: added “(e.g. for some genera, every species from the genera in the region was in trade)”

here theraphosids would suffice, as you indicated above that this taxon refers to tarantulas

Response: edited

you are referring to terrestrial invertebrates, correct, or want to stay general here, then amend accordingly.

Response: some marine species may be targeted for aquaria trade too, but the data does not exist so we have added “terrestrial”

and refer to the CITES trade database with very few listings.

Response: Added

conservation management strategies

Response: edited

Terraristika (two "r") but would add, the largest German reptile fair, the Terraristika"

Response: Thanks for spotting this, added

to improve protection of potentially

Response: thanks, edited

half of the tarantula species scientifically described are traded

Response: edited as suggested

would rather write, " ... 20% of scorpions assigned to the Buthidae...

Response: happy to nod to the challenging taxonomy!

evaluated threatened ...

Response: edited

see my notes on acronyms above

Response: we have left these in full, as not all readers will read results, and we only use the terms a few times

isnt' it rather "and are likely completely unregulated domestically "?

Response: yes, we were trying to be pragmatic, but have edited as suggested, but we added “DD” as it is used later in the paragraph

check CITES notifications, also with regard to quotas, ranches and wild taken

e.g.,

https://www.speciesplus.net/species#/taxon_concepts/5811/legal

Response: Thank you, we have detailed this more in text, it is a fascinating example

what is forms", probably better write "taxa"

Response: changed to “taxa or morphs” (without better data determining which is impossible)

commonly

Response: edited

i.e. species complexes

Response: added

difficult/challenging

Response: changed to challenging

better "locality"

Response: changed

and already in trade

Response: edited

It would be worth considering whether there is not an insider language that also communicates on taxa.

Response: added "or specialist (non-scientific)"

... via social media (Instagram and Facebook)

Response: edited

likely is significantly higher.."

Response: edited

these notes here on many taxonomic issues/ uncertainties need to be also at some point included in the method section, where a section is still missing on taxonomic sources used for this study.

Response: We have added this to methods, lists of synonyms etc are already noted, but we have made the rest of the text clearer

online trade of wild animals is ...

Response: added 'online'

reference and more detail here?

Response: we could not find a reliable reference, so removed this

distinctly exceed

Response: added 'greatly'

been collected in the wild

Response: edited

(monitoring and enforcing these)

Response: added

"..and the verification and registration..."

Response: added

Link to CITES?

Response: yes, added

and source codes (if CITES species are involved) or "as captive-breds"

Response: Added

Edits to methods have also been made as needed

Reviewer #2 (Remarks to the Author):

The authors present research into the global arachnid trade. They look at online trade sites, US imports, and the CITES trade database. They characterize the trade of arachnids by species perform summary analyses about: taxonomic trends, conservation (IUCN) characteristics, year of species discovery vs. entry in trade, country of origin, and more.

It is my assessment that the authors present a valuable contribution that is methodological sound and robust. Their methodology has been used in prior research and is well-documented in the literature. Their results are novel and illuminating on a topic that, to date, has not received much attention. Further, their discussion of results is well presented and not overstated – all the necessary caveats are discussed and woven throughout the paper. Importantly, specific conservation implications are not discussed because their results do not merit such a deep dive, but the authors point to what research needs to be done in order to have specific conservation recommendations. My review is short because I believe this paper is worthy of publication as is. I have read in the manuscript in full and given it careful consideration. I have left some minor comments below that the authors may wish to consider.

I do not wish to remain anonymous and have signed my review.

Oliver C. Stringham

oliver.stringham@adelaide.edu.au

Minor comments:

Line 59: can you give another example other than the Hitler's beetle? Apparently the species was literally named after Hitler, so I don't endorse the common name and would prefer it not to be used where possible. Or perhaps a species of blind cave beetle and leave it at that.

Introduction in general: can you briefly state why arachnids are being traded? Or at least what is known about that? It's not until the discussion that pet trade and consumption are mentioned, but I think it's worth a mention in the introduction.

Response: rephrased to “*known to have nearly driven various species to extinction, especially where niche markets exist (7).*” We also added “*Yet arachnids have become popular pets, often regarded as “cool”, and given that they require little space, making arachnids very practical pets for people without much space, such as many urban settings.*”

Line 135: saying only 2% seems a bit misleading because the maximum number all 3 sources can have in common is 30 (the data source w/least # spp. = # cites spp.), if that makes sense? I would reword to reflect this.

Response: We agree, so have added “*though this is in large part because of how few species are regulated by CITES*”

Line 137: ‘around 100 potential species’ – just say the exact number

Response: we added the exact number and refer to the supplement

Line 148: why the world ‘could’, is it because the true number of species in the group is not known? It wasn't clear (to me) from the text.

Response: removed could

Line 153-154: is this sentence just to caveat you didn't sample every seller/location/etc on earth? To me, it seems out of place or missing a reason why you are cautioning that the number of species in trade may be higher than you observed.

Response: removed

Line 163: what does N.b. mean? Is this a journal specific thing?

Response: Nota benne, from the Latin “note well”

Figure 2: Please mention that one data point represents one taxonomic family. It is evident but it took me a little bit to figure out.

Response: added “, *each point represents one family, with some of the larger families named.*”

Line 181 – 184: I don’t really see that post-2016 trend in the data. 2017 had 500 spp in 133 pages (3.8 spp per page), 2018 had ~400 spp in 280 pages (1.4 spp per page), which seem like a similar rate to the other years. One way to tell for sure is to plot the # spp./page for each year. Further, the number of unique spp for ’17 and ’18 are higher than most other years. For those 2 reasons, the caveat in these lines seems out of place/wrong. Please reconsider this statement.

Response: we removed the caveats

Line 217: the ‘or potentially indicating’ clause doesn’t make sense to me. What is potentially indicating undescribed diversity?

Response: added “*as the consistent use of colloquial names by collectors may represent*”

Figure 4B: Are these colors the literal color descriptions used by sellers? Or was some aggregation used?

Response: added “*(based on actual colour listings).*”

Lines 231 – 234: what to the percentages indicate?

Response: added “*of newly described species*”

Line 240: I think standard deviation is a better metric for variation. Looking at Figure 5, the variation around the mean does not look like 0.4 years – I understand that may be the SE but to me that’s not really useful.

Response: We agree. We have swapped the standard errors for standard deviations in the text and figure for this section, as the variation is more important to highlight than the precision of the reported mean.

Figure 5: can I suggest a discrete color ramp. They tend to me more understandable. In ggplot2 it would be `scale_fill_fermenter()`.

Response: After reviewing what the discrete ramp looks like, we feel that in some areas (particularly South America in 6C), the number of steps required to help interpretation is too high for a clean legend. While we agree that discrete colour ramps can be easier to interpret, we would prefer to keep the continuous gradient as it represents a more direct connection to the data type being presented.

Figure 5: I think the color ramp limits should be the same between D and E. That way they can be directly compared? Just a suggestion.

Response: Because the numbers are so different it is hard to see patterns in smaller groups, or when units vary so this is the clearest with separate ramps

Figure 5f: Are there arachnids in Antarctica?

Response: very few

Figure 5b&c: Why does the US show up as having arachnids traded. I was under the impression you

were solely looking at imports to the US. But perhaps you are also looking at exports out of US? If this is the case, please state this in the main text.

Response: We have double checked the LEMIS data and these are indeed all listed as imports with the origin stated as US. We are limited exploring this much further without additional details from LEMIS, but presume that the instances of USA imports are re-imports. We added “*details of this are featured on online trading forum, and YouTube*” links can be added, but we felt we should maintain peoples anonymity.

Line 299: Also important to note that these individuals could have been ‘re exported’ meaning they entered the non-native country and then were exported at a later time after spending some time in the country. Or are captively bred and then exported as wild caught, etc. We talk about this a little in: Sinclair et al., The International Vertebrate Pet Trade Network and Insights from US Imports of Exotic Pets, BioScience, Volume 71, Issue 9, September 2021, Pages 977–990, <https://doi.org/10.1093/biosci/biab056>

Response: Thanks, we added “, *and some species may have been re-exported*”, we also now touch on it in discussion

Line 362: any references for this? Or was this something you observed in the advertisements you saw? Please specify.

Response: Observed, we added text to reflect this

Line 369: The link did not work for me. Perhaps worth taking a screen shot and putting in Appendix so that there already be a record of what you are referring to.

Response: An appendix has been added

Line 375: we discuss the different types of websites and the limitations of web data for wildlife trade, if it is a useful reference here, feel free to use it: Stringham, et al. (2021), A guide to using the internet to monitor and quantify the wildlife trade. Conservation Biology, 35: 1130-1139. <https://doi.org/10.1111/cobi.13675>

Response: Thanks! This is very useful and has been added

line 399: CITES doesn’t handle domestic trade. I don’t think you were saying this here but it was a little ambiguous (in my opinion)

Response: Thanks, edited to ‘as they are rarely covered by domestic policy.’

line 416: we noted this in the above-mention Sinclair paper for vertebrates so it’s a problem not just limited to arachnids. Perhaps a wildlife/pet trade problem in general

line 439: what does cf. mean? Sorry for my ignorance but I haven’t come across this

Response: We’ve added the following to make it much easier to follow “(which can both denote that the putative species has not been formally described)”

lines 449 to 451: very nicely worded

Response: Thank you!

line 498: it is unclear (to me) how the use of junior synonyms results in not effective regulation. I don’t get the link. Perhaps ineffective enforcement, if you believe the use of junior synonyms means that the true species won’t get flagged. I think the US system should (although I don’t really know)

have a cross reference to taxonomic database. First, these boxes would only be at the seller level, not at the import/export level, right? So presumably the actual species/individuals will still be subject to country-border level regulations.

Response: we've added "*Junior synonyms can also mean that a species which is subject to CITES or other regulations may continue to be traded under a junior synonym or old name, and thus newer legislation is likely not to be applied when the species is traded*"

Second, I don't think the mystery boxes necessarily mean anything bad. It could mean store just wanted to get rid of common stock. This might be an optimistic take but I don't see the benefit to sell desired/rare species this way because they would get more money if they sold them normally. Unless you know more specifics on it or have a reference, I would remove this part.

Response: We added "*as not only is it impossible to monitor what is being traded, but they could be use to facilitate laundering of protected species.*"

Line 520: This sentence is unclear to me. What exactly is a trade-blocs/area in this context? A country?

Response: Thanks, we have added "*(facilitate regional trade of captive bred individuals, and heighten regulations of external trade)*"

Line 531: It's worth noting that many of these regulation suggestions require effective enforcement. And that enforcement is difficult. This is touched on Cardoso's and Fukushima's recent synthesis papers.

Response: Thanks, now noted

Line 541: what are these lessons? Left me on a cliff-hanger. I think it's worth mentioning one of them.

Response: rephrased to make clear

Line 562: Lack of seizures doesn't mean that illegal species aren't passing through USFWS surveillance. We observed they do occasionally let 'illegal' species in without seizing them. For example, we noted this for an Australian endemic reptile: Heinrich, et al. (2022), Strengthening protection of endemic wildlife threatened by the international pet trade: The case of the Australian shingleback lizard. *Anim. Conserv.*, 25: 91-100. <https://doi.org/10.1111/acv.12721> . I would guess this is even more of a problem for invertebrate if it's a problem for a highly recognizable vertebrate species.

Response: Now noted

Line 572: I know our papers came out around the same time but the above Stringham et al. (2021) outlines a procedure for monitoring the web for wildlife trade. If you feel it's appropriate, you can cite it here as well.

Response: Thanks, we now note it in discussion

Methods: worth mentioning you did not look at social media/social media groups.

Great paper, well done!

Response: Thank you, and for your useful comments

Reviewer #3 (Remarks to the Author):

I found this manuscript well written and the analyses well done. The actual content of the paper is novel, well researched and greatly brings to the forefront the rapidly growing plight in world trade of targeted invertebrates.

Response: Thank you, greatly appreciated!

My only criticism would be that I would have liked to see 'Discussion' follow 'Methods', rather than the way the paper is presently laid out. I think categories like 'Conclusion' or the final 'Discussion' should follow all the methods that were used to reach those categories.

Response: Thanks, we will follow the journals preference on this, it was in the “Nature family” format which generally has this order

The remainder of my comments that need attention are found in the returned ms, however, once addresses, they should not impede the publication of this ms.

Sincerely, Rick C. West

Add comma after ‘trade’

Response: done

P2. Remove comma (x3)

Response: done

P9. Remove comma (x3)

Response: done

P11. Remove comma (x2)

Response: done

Add space

Response: Done

Aphonopelma seemani- add ‘n’

Response: Done

P13

When first mentioning a species, provide author and date - *Grammostola pulchra* Mello-Leitao 1921. [Please provide accent over 'a' in Leitao. I could not do that in this comment format]

Response: one

As above. When first listing a species, provide author and date - Montes de Oca, D'Elia & Perez-Miles 2016.

[Please check and provide accents over the 'i' in Elia and the first 'e' in Perez ... I could not do that in this comment format.]

Response: Done

Provide author and date - *Dolichothele diamantinensis* Revollo, Silva & Bertani 2017.

Response: Done

P15. Delete hyphen and space between species and authors. Ideally, add all the authors names so you're consistent throughout this paper when first listing a species. Delete comma between authors and date.

Response: Done

Delete superscript bracket and replace with proper font bracket.

Response: Thank you for spotting this, now fixed

Delete '78' superscript and replace with reference this number in regular italics font brackets.

Response: Thanks, corrected

In italics (x2)

Response: Done

pers. comm., and add the year after the pers. comm.

Response: Added 2021

Reference comments

Response: References have been checked to ensure consistency with published journal articles, this includes the use of “et al” for papers with 5 or more authors. Spelling and italics, as well as reference to access dates have been corrected

**** See the Nature Portfolio author and referees' website at www.nature.com/authors for information about policies, services and author benefits**

Communications Biology is committed to improving transparency in authorship. As part of our efforts in this direction, we are now requesting that all authors identified as ‘corresponding author’ create and link their Open Researcher and Contributor Identifier (ORCID) with their account on the Manuscript Tracking System prior to acceptance. ORCID helps the scientific community achieve unambiguous attribution of all scholarly contributions. You can create and link your ORCID from the home page of the Manuscript Tracking System by clicking on ‘Modify my Springer Nature account’ and following the instructions in the link below. Please also inform all co-authors that they can add their ORCIDs to their accounts and that they must do so prior to acceptance.

If you experience problems in linking your ORCID, please contact the Platform Support Helpdesk.

This email has been sent through the Springer Nature Tracking System NY-610A-NPG&MTS

Confidentiality Statement:

This e-mail is confidential and subject to copyright. Any unauthorised use or disclosure of its contents is prohibited. If you have received this email in error please notify our Manuscript Tracking System Helpdesk team at <http://platformsupport.nature.com>.

Details of the confidentiality and pre-publicity policy may be found here <http://www.nature.com/authors/policies/confidentiality.html>

Privacy Policy | Update Profile